# Subacute SARS-CoV-2 replication can be controlled in the absence of CD8+ T cells in cynomolgus macaques

Takushi Nomura[1]*, Hiroyuki Yamamoto[1], Masako Nishizawa[1], Trang Thi Thu Hau[1], Shigeyoshi Harada[1], Hiroshi Ishii[1], Sayuri Seki[1], Midori Nakamura-Hoshi[1], Midori Okazaki[1], Sachie Daigen[1], Ai Kawana-Tachikawa[1,2,3], Noriyo Nagata[4], Naoko Iwata-Yoshikawa[4], Nozomi Shiwa[4], Shun Iida[4], Harutaka Katano[4], Tadaki Suzuki[4], Eun-Sil Park[5], Ken Maeda[5], Yuriko Suzaki[6], Yasushi Ami[6], Tetsuro Matano[1,2,3]*

1 AIDS Research Center, National Institute of Infectious Diseases, Tokyo, Japan, 2 Institute of Medical Science, University of Tokyo, Tokyo, Japan, 3 Joint Research Center for Human Retrovirus Infection, Kumamoto University, Kumamoto, Japan, 4 Department of Pathology, National Institute of Infectious Diseases, Tokyo, Japan, 5 Department of Veterinary Science, National Institute of Infectious Diseases, Tokyo, Japan, 6 Management Department of Biosafety, Laboratory Animal, and Pathogen Bank, National Institute of Infectious Diseases, Tokyo, Japan

* nomutaku@nih.go.jp (TN); tmatano@nih.go.jp (TM)

**Data Availability Statement:** All relevant data are within the manuscript and its Supporting Information files.

## Abstract

SARS-CoV-2 infection presents clinical manifestations ranging from asymptomatic to fatal respiratory failure. Despite the induction of functional SARS-CoV-2-specific CD8[+] T-cell responses in convalescent individuals, the role of virus-specific CD8[+] T-cell responses in the control of SARS-CoV-2 replication remains unknown. In the present study, we show that subacute SARS-CoV-2 replication can be controlled in the absence of CD8[+] T cells in cynomolgus macaques. Eight macaques were intranasally inoculated with $10^5$ or $10^6$ $TCID_{50}$ of SARS-CoV-2, and three of the eight macaques were treated with a monoclonal anti-CD8 antibody on days 5 and 7 post-infection. In these three macaques, CD8[+] T cells were undetectable on day 7 and thereafter, while virus-specific CD8[+] T-cell responses were induced in the remaining five untreated animals. Viral RNA was detected in nasopharyngeal swabs for 10–17 days post-infection in all macaques, and the kinetics of viral RNA levels in pharyngeal swabs and plasma neutralizing antibody titers were comparable between the anti-CD8 antibody treated and untreated animals. SARS-CoV-2 RNA was detected in the pharyngeal mucosa and/or retropharyngeal lymph node obtained at necropsy on day 21 in two of the untreated group but undetectable in all macaques treated with anti-CD8 antibody. CD8[+] T-cell responses may contribute to viral control in SARS-CoV-2 infection, but our results indicate possible containment of subacute viral replication in the absence of CD8[+] T cells, implying that CD8[+] T-cell dysfunction may not solely lead to viral control failure.

**Funding:** This study was supported by Research Programs from Japan Agency for Medical Research and Development (AMED) (JP19fk0108104 to AK-T; JP20nk0101601, JP21fk0410035, JP21fk0108125, JP21jk0210002 to TM) and by JSPS Grants-in-Aid for Scientific Research from Ministry of Education, Culture, Sports, Science and Technology (MEXT) in Japan (21H02745 to TM]). The funders had no role in study design, data collection and analysis, decision to publish, or preparation of the manuscript.

**Competing interests:** The authors have declared that no competing interests exist.

## Author summary

SARS-CoV-2 infection presents a wide spectrum of clinical manifestations ranging from asymptomatic to fatal respiratory failure. The determinants for failure in viral control and/or fatal disease progression have not been elucidated fully. Both acquired immune effectors, antibodies and CD8$^+$ T cells, are considered to contribute to viral control. However, it remains unknown whether a deficiency in either of these two arms is directly linked to failure in the control of SARS-CoV-2 replication. In the present study, to know the requirement of CD8$^+$ T cells for viral control after the establishment of infection, we examined the effect of CD8$^+$ cell depletion by monoclonal anti-CD8 antibody administration in the subacute phase on SARS-CoV-2 replication in cynomolgus macaques. Unexpectedly, our analysis revealed no significant impact of CD8$^+$ cell depletion on viral replication, indicating that subacute SARS-CoV-2 replication can be controlled in the absence of CD8$^+$ T cells. CD8$^+$ T-cell responses may contribute to viral control in SARS-CoV-2 infection, but this study suggests that CD8$^+$ T-cell dysfunction may not solely lead to viral control failure or fatal disease progression.

## Introduction

The coronavirus disease 2019 (COVID-19) caused by severe acute respiratory syndrome coronavirus 2 (SARS-CoV-2) has rapidly spread resulting in a major pandemic [1]. SARS-CoV-2 transmission occurs via the respiratory route, and the average incubation period from infection to symptom onset has been estimated to be 5 days [2]. SARS-CoV-2 infection presents a wide spectrum of clinical manifestations ranging from asymptomatic to fatal respiratory failure [3]. Multiple cofounding factors such as age and underlying diseases are associated with COVID-19 severity [4–8]. For instance, auto-antibodies against type I interferon have been reported to be associated with life-threatening COVID-19 pneumonia [9,10]. However, the exact determinants for failure in viral control and/or fatal disease progression have not been elucidated fully.

Most non-fatal COVID-19 cases show a limited period of detectable virus production in pharyngeal swabs peaking at around one week post-infection [11]. Host acquired as well as innate immune responses are involved in the control of viral replication [8,12–14]. Anti-SARS-CoV-2 neutralizing antibodies are induced in most infected individuals [13–16]. Recent clinical studies including those on convalescent plasma and/or monoclonal antibody administration have indicated efficacy of neutralizing antibodies against SARS-CoV-2 infection [17–20]. Animal studies have confirmed *in vivo* efficacy of neutralizing antibodies against infection [21–25]. Also, SARS-CoV-2-specific T-cell responses are induced in most non-fatal COVID-19 cases [26–28]. Current studies have indicated induction of functional virus-specific CD8$^+$ T-cell responses in convalescent COVID-19 individuals, implying suppressive pressure of CD8$^+$ T cells on SARS-CoV-2 replication [29,30]. Thus, both acquired immune effectors, antibodies and CD8$^+$ T cells, are considered to contribute to viral control. However, it remains unknown whether a deficiency in either of these two arms is directly linked to failure in the control of SARS-CoV-2 replication. It has been reported that COVID-19 patients with agammaglobulinemia controlled disease progression, suggesting viral control even in the absence of antibody responses [31].

A previous study of anti-CD8 antibody administration prior to re-infection in rhesus macaques has indicated partial contribution of CD8$^+$ T cells to protection against SARS-CoV-2 re-infection [25]. However, the requirement of CD8$^+$ T cells for the control of virus replication after the establishment of infection remains unclear. In the present study, we investigated

the effect of CD8[+] cell depletion by monoclonal anti-CD8 antibody administration in the sub-acute phase on SARS-CoV-2 replication in cynomolgus macaques. Unexpectedly, our analysis revealed no significant impact of CD8[+] cell depletion on viral replication, indicating that sub-acute SARS-CoV-2 replication can be controlled in the absence of CD8[+] T cells.

## Results

### Kinetics of SARS-CoV-2 infection in cynomolgus macaques after intranasal inoculation

Previous studies have shown that intranasal and intratracheal inoculation with $10^5$ TCID$_{50}$ (50% tissue culture infective doses) of SARS-CoV-2 results in the establishment of infection in rhesus macaques, with viral RNA detectable for more than a week post-infection in pharyngeal swabs [32,33]. In the present study, we first examined whether intranasal SARS-CoV-2 inoculation only can result in viral infection in cynomolgus macaques. In the first experiment, cynomolgus macaques were intranasally inoculated with $10^6$ (exactly 7.5 x $10^5$ in macaque N011), $10^5$ (exactly 7.5 x $10^4$ in macaques N012 and N013), or $10^4$ (exactly 7.5 x $10^3$ in macaque N014) TCID$_{50}$ of SARS-CoV-2 (Table 1). Macaques N011, N012, and N013 showed similar levels of viral RNA in nasopharyngeal swabs on day 2, at the peak (Fig 1A). Viral RNA was also detected in throat swabs with a lower peak (Fig 1B). Viral RNA in nasopharyngeal swabs was detectable for approximately two weeks (up to: day 17 in N011, day 12 in N012, and day 14 in N013) after virus inoculation (Fig 1A, 1C and 1D). Subgenomic RNAs (sgRNAs) were also detected in nasopharyngeal and throat swabs, indicating viral replication (Fig 2A and 2B). SARS-CoV-2 sgRNAs were detected in nasopharyngeal swabs until day 9 in N011, day 7 in N012, and day 5 in N013 (Fig 2A, 2C and 2D). However, in macaque N014, which was inoculated with $10^4$ TCID$_{50}$ of SARS-CoV-2, sgRNAs were undetectable, and viral RNAs were detectable albeit at lower levels, only until day 5 in nasopharyngeal swabs (Figs 1A and 2A), indicating that $10^4$ TCID$_{50}$ is below the virus inoculum threshold to consistently induce detectable viral replication. N014 was subsequently excluded from further analyses.

In the second experiment, two (N021 and D023) and three (N022, D024, and D025) macaques were intranasally inoculated with $10^6$ and $10^5$ TCID$_{50}$ of SARS-CoV-2, respectively (Table 1). Monoclonal anti-CD8 antibody was administered intravenously on days 5 and 7 to three (D023, D024, and D025 in Group D) of the five macaques. All of the five macaques in the second experiment showed comparable levels of viral RNAs and sgRNAs in nasopharyngeal swabs on day 2 compared to the three macaques inoculated with $10^6$ or $10^5$ TCID$_{50}$ of SARS-CoV-2 in the first experiment (Figs 1A and 2A). Indeed, no significant difference in RNA levels in nasopharyngeal swabs on day 5 was observed between the first three and the second five animals (Fig 1E). No clear difference in viral loads in either nasopharyngeal or throat swabs on days 2 and 5 was observed between macaques inoculated with $10^6$ (n = 3) and $10^5$ (n = 5) TCID$_{50}$ of SARS-CoV-2 (Figs 1C, 1D, 2C and 2D). Viral RNA in nasopharyngeal swabs was detectable until day 14 in N021 and day 10 in N022 following inoculation (Fig 1A, 1C and 1D). SARS-CoV-2 sgRNAs in nasopharyngeal swabs were detected until day 5 in N021 and day 7 in N022 following inoculation (Fig 2A, 2C and 2D). Collectively, in the first and second experiments, intranasal inoculation of cynomolgus macaques with $10^6$ or $10^5$ TCID$_{50}$ of SARS-CoV-2 resulted in viral replication with viral RNA detectable for 10–17 days in nasopharyngeal swabs.

### Kinetics of SARS-CoV-2 infection after CD8[+] cell depletion

We then investigated the effect of CD8[+] cell depletion on viral replication in the subacute phase of SARS-CoV-2 infection. In the three Group D macaques administered with anti-CD8

**Table 1. Macaque experimental protocol.**

| Group | Experiment[a] | Macaques | Gender | Age (yrs) | SARS-CoV-2 dose[b] ($TCID_{50}$) | anti-CD8 Ab Tx[c] | Necropsy[d] |
|---|---|---|---|---|---|---|---|
| N | 1 | N011 | male | 6 | $10^6$ | NT | d21 |
| N | 1 | N012 | male | 6 | $10^5$ | NT | d21 |
| N | 1 | N013 | male | 6 | $10^5$ | NT | d21 |
| _[e] | 1 | N014 | male | 6 | $10^4$ | NT | d21 |
| N | 2 | N021 | female | 3 | $10^6$ | NT | d14 |
| N | 2 | N022 | female | 3 | $10^5$ | NT | d21 |
| D | 2 | D023 | male | 6 | $10^6$ | d5 & d7 | d21 |
| D | 2 | D024 | male | 6 | $10^5$ | d5 & d7 | d21 |
| D | 2 | D025 | female | 3 | $10^5$ | d5 & d7 | d21 |

[a]Two sets of experiments were performed using the same SARS-CoV-2 inoculum stock.

[b]Macaques were intranasally inoculated with the indicated doses ($10^6$ [exactly 7.5 x $10^5$], $10^5$ [exactly 7.5 x $10^4$], or $10^4$ [exactly 7.5 x $10^3$] $TCID_{50}$) of SARS-CoV-2 on day 0.

[c]Macaques in Group D were treated intravenously with anti-CD8 antibody on days 5 and 7 post-infection. NT, not treated.

[d]Macaques were euthanized and necropsied on day 14 or 21 post-infection.

[e]N014 was excluded from comparisons between groups N and D.

antibody on days 5 and 7, CD8$^+$ T cells were undetectable in peripheral blood on day 7 and thereafter (Fig 3). These three macaques showed comparable levels of viral RNA in nasopharyngeal swabs before (day 5) and after (day 7) anti-CD8 antibody treatment compared to the five untreated Group N macaques (Fig 1F). Viral RNA in nasopharyngeal swabs was detectable until day 10 in D025 and day 14 in macaques D023 in D024 after virus inoculation (Fig 1A, 1C and 1D). Viral sgRNAs in nasopharyngeal swabs were detected until day 2 in D023, day 5 in D024, and day 7 in D025 (Fig 2A, 2C and 2D). Collectively, no clear difference in viral RNA levels in swabs was observed for the three anti-CD8 antibody-treated Group D versus the five untreated Group N macaques.

We also examined whether virus could be recovered from individual swab samples (Table 2). SARS-CoV-2 was recovered from nasopharyngeal and throat swabs from all eight animals intranasally inoculated with either $10^6$ or $10^5$ $TCID_{50}$. Virus was recovered for 2–12 days in anti-CD8 antibody-untreated macaques (until day 2 in N021, day 5 in N013 and N022, day 7 in N011, and day 12 in N012) and for 2–7 days in anti-CD8 antibody-treated macaques (until day 2 in D024 and day 7 in D023 and D025). There was no indication of enhanced virus recovery after CD8 cell depletion.

Macaque N021 was euthanized on day 14, while the remaining animals were euthanized on day 21 post-infection (Table 1). Examination of body temperature showed transient slight fever in some animals (on day 2 in N021 and D025; on day 6 in D023 and D024; on days 13–19 in N012) (S1 Fig). Histopathological analysis of the lung obtained at necropsy on day 14 in macaque N021 revealed mild or moderate pulmonary inflammation (S2 Fig), whereas no significant pathology in the lung was detected on day 21 in other animals.

RNA was extracted from the pharyngeal mucosa, retropharyngeal lymph nodes (RPLN), lung, intestine, and spleen obtained at necropsy, and subjected to RT-PCR for detection of viral RNA (Table 3). Viral RNA was undetectable in tissues from macaques N012, N014, N021, D023, and D024. However, viral RNA was detected in the RPLN of N011, in the pharyngeal mucosa, RPLN, and spleen of N013, and in the spleens of N022 and D025. Additionally, viral sgRNAs were also detectable in pharyngeal mucosa, RPLN, and spleen of N013. There was no evidence of enhanced viral replication in anti-CD8 antibody-treated macaques.

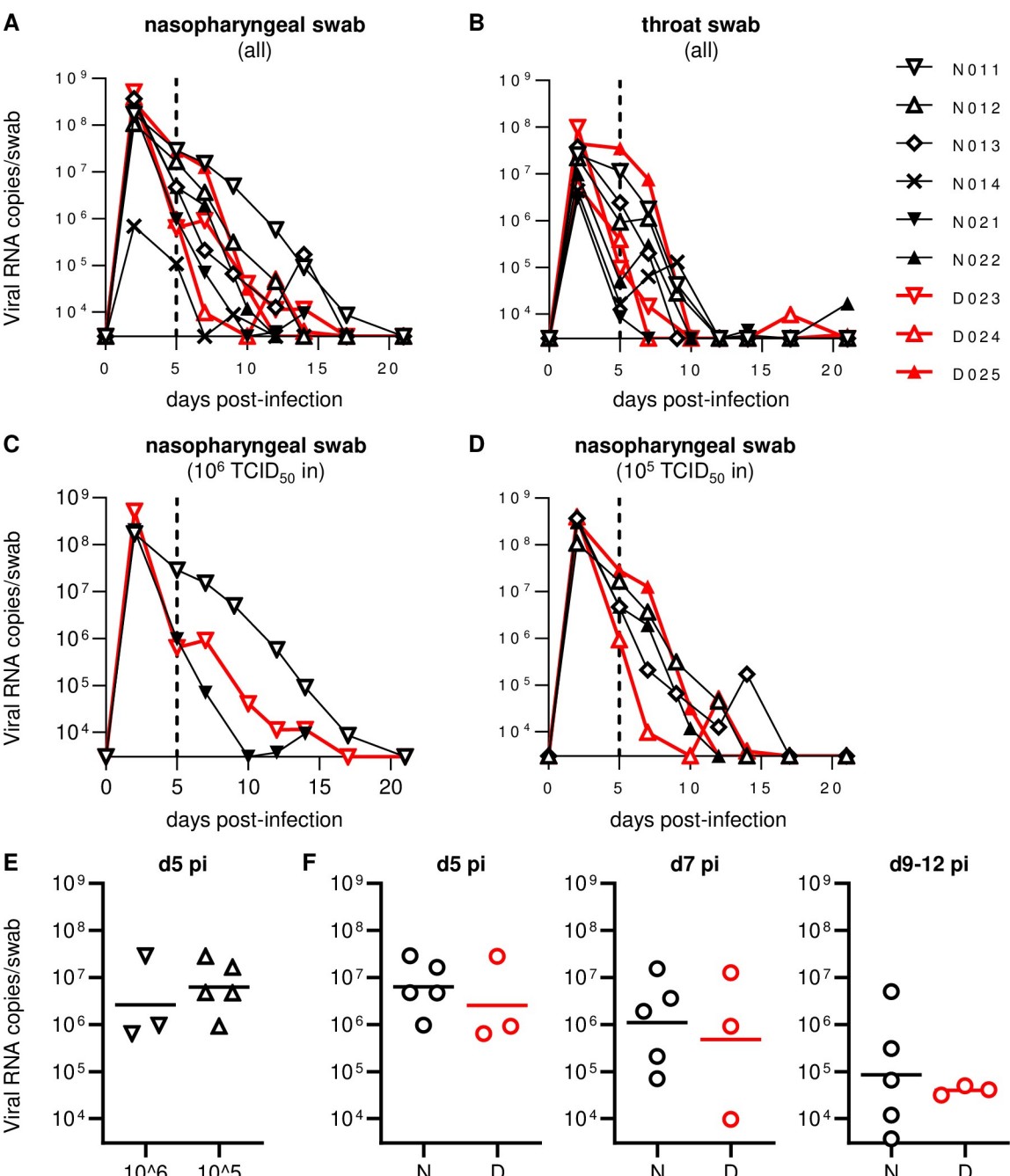

**Fig 1. Viral RNA levels in swabs.** (A-D) Changes in viral RNA levels in nasopharyngeal (A, C, D) and throat (B) swabs after SARS-CoV-2 infection in all animals (A, B) or those infected with $10^6$ (C) or $10^5$ (D) $TCID_{50}$ of SARS-CoV-2. The lower limit of detection was approximately $3 \times 10^3$ copies/swab. (E) Comparison of viral RNA levels in nasopharyngeal swabs at day 5 post-infection between $10^6$ $TCID_{50}$-infected and $10^5$ $TCID_{50}$-infected macaques. No significant difference was observed. (F) Comparison of viral RNA levels in nasopharyngeal swabs at days 5 (left), 7 (middle), and 9–12 (right) post-infection between Group N and D animals infected with $10^6$ or $10^5$ $TCID_{50}$ of SARS-CoV-2. No significant difference was observed.

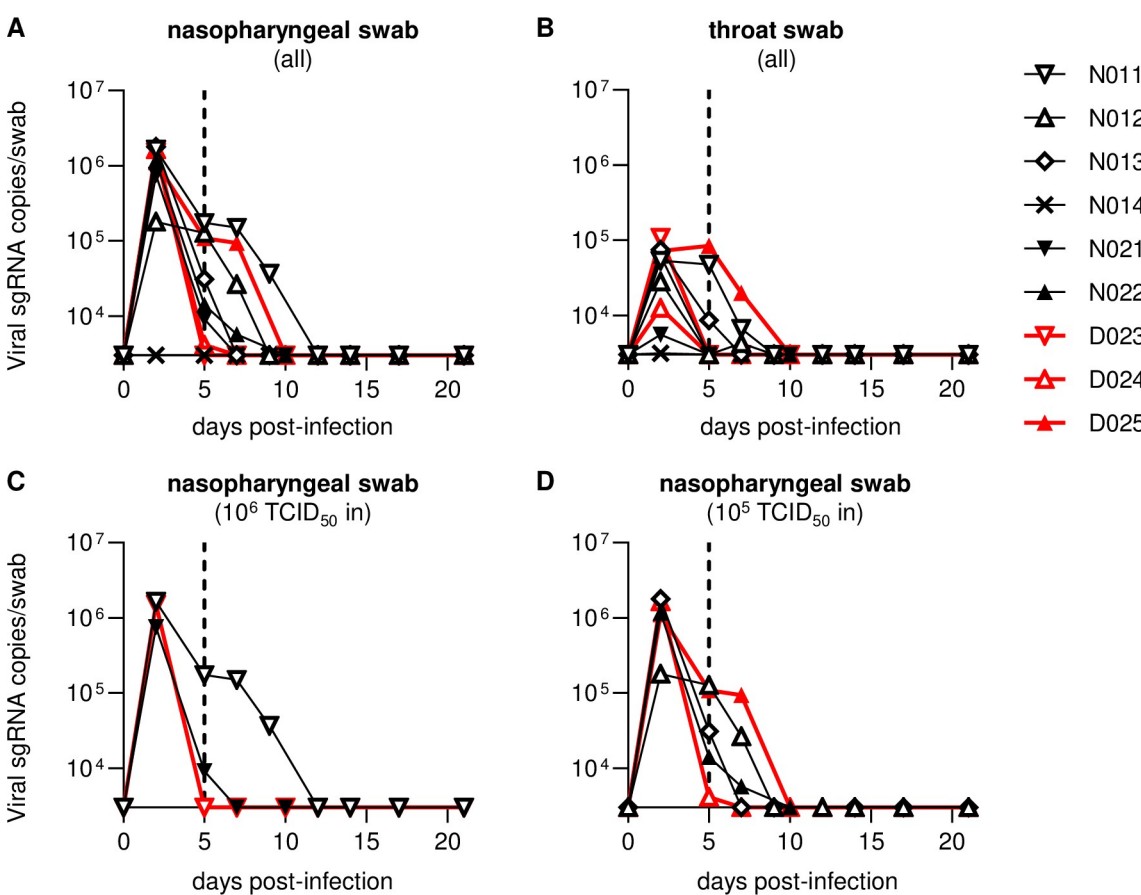

**Fig 2. Viral subgenomic RNA levels in swabs.** Changes in viral sgRNA levels in nasopharyngeal (A, C, D) and throat (B) swabs after SARS-CoV-2 infection in all animals (A, B) or those infected with $10^6$ (C) or $10^5$ (D) $TCID_{50}$ of SARS-CoV-2. The lower limit of detection was approximately 3 x $10^3$ copies/swab.

## Antibody and T-cell responses in macaques after intranasal SARS-CoV-2 inoculation

Anti-SARS-CoV-2 neutralizing antibody (NAb) responses were induced in all the macaques after intranasal SARS-CoV-2 inoculation (Fig 4) NAb responses were detected on day 7 in macaques N021 and D025 only, and in all animals on day 14. Macaques D025 and D024 exhibited the highest and lowest NAb titers, respectively. No clear difference in NAb responses was observed between the three anti-CD8 antibody-treated and the five untreated macaques.

Finally, we examined CD8+ T-cell responses specific for SARS-CoV-2 spike (S), nucleocapsid (N), and membrane-and-envelope (M&E) antigens in the five anti-CD8 antibody-untreated macaques inoculated with $10^6$ or $10^5$ $TCID_{50}$ of SARS-CoV-2. In the analysis using peripheral blood mononuclear cells (PBMCs), SARS-CoV-2-specific CD8+ T-cell responses were undetectable in macaque N013 but detected in the remaining four macaques (Fig 5A and 5B). Macaque N022 exhibited CD8+ T-cell responses on day 7 while the remaining three macaques (N011, N012, and N021) showed initial SARS-CoV-2 specific responses on day 14. Analysis using submandibular lymph nodes (SMLN) obtained at necropsy found SARS-CoV-2-specific CD8+ T-cell responses in macaques N011, N013, and N021 (Fig 5C). Interestingly, SARS-CoV-2-specific CD8+ T-cell responses were undetectable in PBMCs but detected in SMLN in macaque N013. Thus, SARS-CoV-2-specific CD8+ T-cell responses were detected in

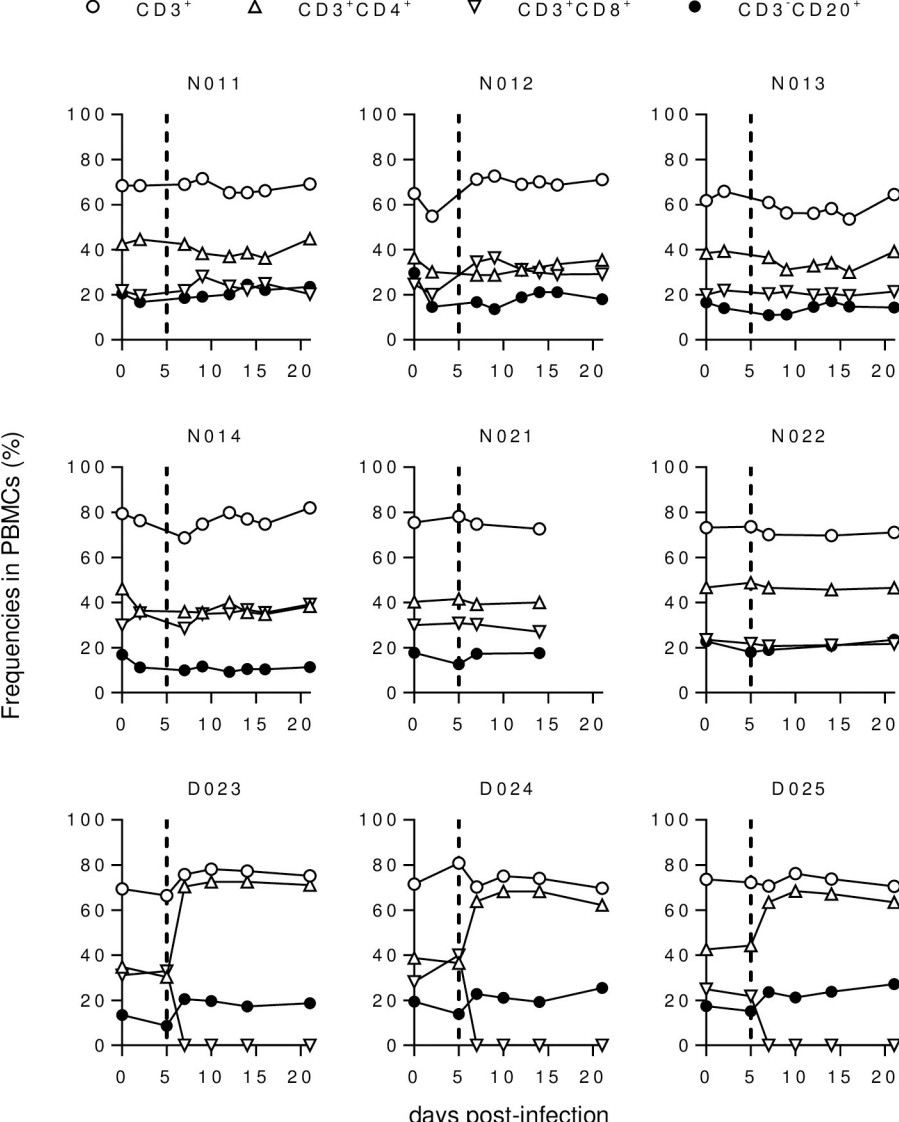

**Fig 3. Peripheral blood B- and T-cell frequencies.** Changes in %CD3$^+$, %CD3$^+$CD4$^+$, %CD3$^+$CD8$^+$, and % CD3$^-$CD20$^+$ T cells in macaque PBMCs after SARS-CoV-2 infection.

all the five anti-CD8 antibody-untreated Group N macaques inoculated with 10$^6$ or 10$^5$ TCID$_{50}$ of SARS-CoV-2. Finally, CD8$^+$ T-cell depletion was confirmed in SMLN obtained at necropsy from the anti-CD8 antibody-treated macaques (S3 Fig).

## Discussion

Host T-cell and B-cell responses have been reported to contribute to the control of SARS-CoV-2 replication [8,12,13,28]. In a murine model of infection with a mouse-adapted strain of SARS-CoV, depletion of CD4$^+$ T cells resulted in reduced neutralizing antibody responses and delayed virus clearance from the lung [34]. Furthermore, SARS-CoV replication was controlled in the absence of CD4$^+$ T and B cells, implicating CD8$^+$ T cells in viral control [35]. Recent studies in humans have shown that functional virus-specific CD8$^+$ T-cell responses are induced in convalescent COVID-19 individuals [29,30]. These reports suggest contribution of

**Table 2. Virus recovery from pharyngeal swabs after SARS-CoV-2 infection.**

| Macaques | Virus recovery from swabs[a] | | | | | | | | |
|---|---|---|---|---|---|---|---|---|---|
| | d0 | d2 | d5 | d7 | d9/10 | d12 | d14 | d17 | d21 |
| N011 | - | + | + | + | - | - | - | - | - |
| | - | + | + | - | - | - | - | - | - |
| N012 | - | + | + | + | - | + | - | - | - |
| | - | + | - | - | - | - | - | - | - |
| N013 | - | + | - | - | - | - | - | - | - |
| | - | + | + | - | - | - | - | - | - |
| N014 | - | - | - | - | - | - | - | - | - |
| | - | + | - | - | - | - | - | - | - |
| N021 | - | + | - | - | - | - | - | - | - |
| | - | + | - | - | - | - | - | - | - |
| N022 | - | + | + | - | - | - | - | - | - |
| | - | + | - | - | - | - | - | - | - |
| D023 | - | + | - | + | - | - | - | - | - |
| | - | + | - | - | - | - | - | - | - |
| D024 | - | + | - | - | - | - | - | - | - |
| | - | + | - | - | - | - | - | - | - |
| D025 | - | + | + | + | - | - | - | - | - |
| | - | + | + | - | - | - | - | - | - |

[a]Swab samples were added to Vero E6/TMPRSS2 cell culture to recover infectious virus. + indicates successful virus recovery from nasopharyngeal (upper row) or throat (lower row) swabs for each animal.

CD8[+] T cells in the control of SARS-CoV-2 replication. However, it remains unclear whether SARS-CoV-2 replication can be controlled in the absence of CD8[+] T cells. In the present study, we investigated the impact of depletion of CD8[+] cells (including CD8[+] T cells) by anti-CD8 antibody administration on SARS-CoV-2 replication in the subacute phase after establishment of virus infection. Our results on viral RNA and virus recovery from pharyngeal swabs (Fig 2 and Table 2) indicate that viral replication was not contained when animals were treated with anti-CD8 antibody, while viral replication was controlled after the CD8[+] cell depletion. We found no significant enhancement of viral replication or delay in viral clearance

**Table 3. Detection of viral RNA in tissues obtained at necropsy.**

| Macaques | Autopsy | Detection of viral RNA in tissues[a] | | | | |
|---|---|---|---|---|---|---|
| | | Pharyngeal mucosa | Retropharyngeal lymph node | Lung | Intestine | Spleen |
| N011 | d21 | - | + | - | - | - |
| N012 | d21 | - | - | - | - | - |
| N013 | d21 | + | + | - | - | + |
| N014 | d21 | - | - | - | - | - |
| N021 | d14 | - | - | - | - | - |
| N022 | d21 | - | - | - | - | + |
| D023 | d21 | - | - | - | - | - |
| D024 | d21 | - | - | - | - | - |
| D025 | d21 | - | - | - | - | + |

[a]RNA was extracted from individual tissues and subjected to RT-PCR to detect SARS-CoV-2 RNA. + indicates detection of viral RNA.

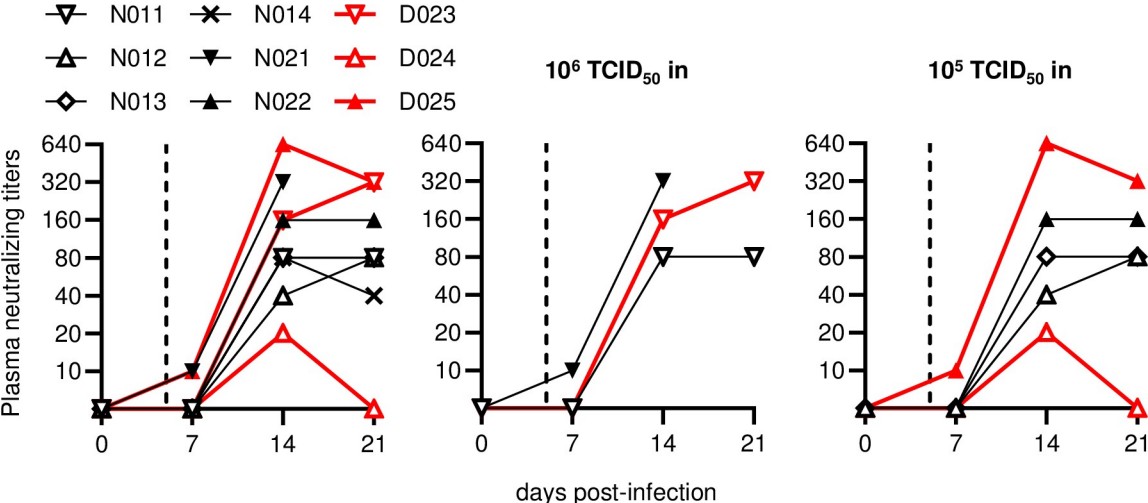

**Fig 4. SARS-CoV-2-specific neutralizing antibody responses.** Changes in plasma anti-SARS-CoV-2 neutralizing antibody titers post-infection in all animals (left) or those infected with $10^6$ (middle) or $10^5$ (right) TCID$_{50}$ of SARS-CoV-2.

after CD8$^+$ cell depletion, indicating that subacute SARS-CoV-2 replication can be controlled in the absence of CD8$^+$ T cells.

Our findings do not deny the contribution of CD8$^+$ T cells in the control of SARS-CoV-2 replication or the possibility of viral protection by vaccine-induced CD8$^+$ T cells. Virus-specific CD8$^+$ T-cell responses were mostly undetectable at week 1 and became detectable at week 2 post-infection in the present study, which is consistent with a recent report on T-cell responses in the acute phase after onset in COVID-19 patients [29]. Thus, CD8$^+$ T-cell responses may not play a central role in the control of peak viral load but could have a large impact on the containment of viral replication and/or disease progression after that in primary SARS-CoV-2 infection. Contribution of CD8$^+$ T-cell responses to protection against re-infection has been suggested [25], implying that vaccine-induced CD8$^+$ T-cell responses may enhance viral control in the acute phase. What is indicated in the present study is that CD8$^+$ T-cell dysfunction is not directly linked to failure in viral control, possibly implying that there may be multiple arms of host immune mechanisms involved in containing primary SARS-CoV-2 replication.

An animal model for SARS-CoV-2 infection is necessary for analysis of pathogenesis and transmission and the evaluation of vaccines and anti-viral drugs. Non-human primate models are recognized as being the most clinically relevant because of their genetic and physiological similarities to humans. Recent studies have shown that rhesus and cynomolgus macaques can be infected with SARS-CoV-2 and exhibit clinical manifestations resembling human COVID-19 [32,36–38]. Both macaque species present mild to moderate forms of COVID-19, which is observed in the majority of the human population. We thus used a model of SARS-CoV-2 infection in cynomolgus macaques for analysis of the effect of CD8$^+$ cell depletion on virus replication.

We attempted SARS-CoV-2 inoculation via the intranasal route only without intratracheal inoculation, because it may more closely reflect viral transmissions in humans. The geometric means of peak viral RNAs and sgRNAs in nasopharyngeal swabs (on day 2) were 2.7 x $10^8$ (range: 1.1 x $10^8$ to 5.2 x $10^8$) and 1.0 x $10^6$ (range: 1.8 x $10^5$ to 1.8 x $10^6$) copies/swab, respectively, which are equivalent to those in rhesus macaques inoculated both intranasally and intra-tracheally with $10^5$ TCID$_{50}$ of SARS-CoV-2 [32,33]. In macaques inoculated with $10^5$ TCID$_{50}$

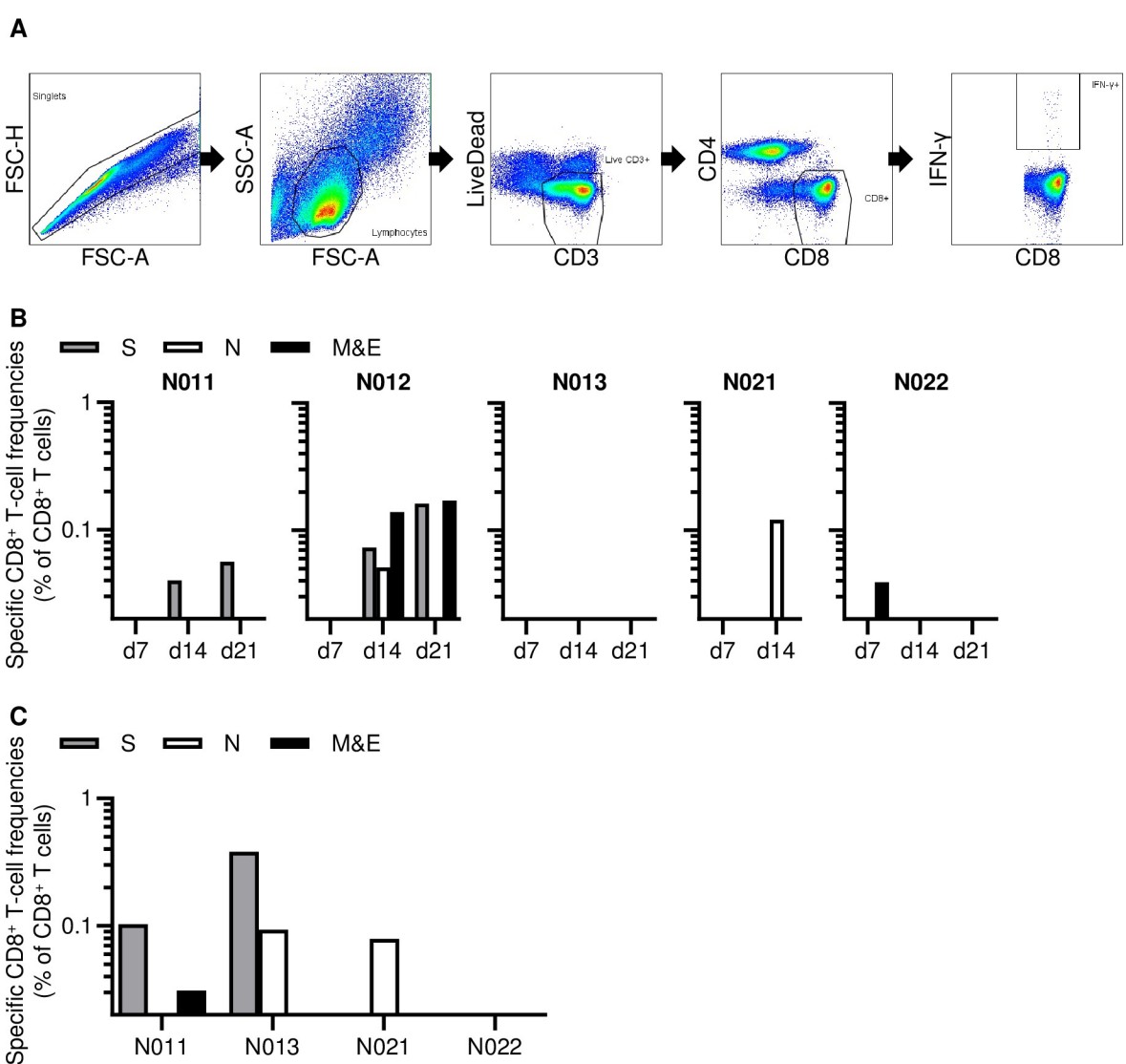

**Fig 5. SARS-CoV-2-specific CD8$^+$ T-cell responses.** (A) Representative gating schema for detection of IFN-γ induction after stimulation with overlapping M&E peptide pools in macaque N012 on day 14 post-infection. (B) Frequencies of CD8$^+$ T cells targeting S, N, and M&E in PBMCs on days 7, 14, and 21 post-infection in Group N animals infected with $10^6$ (middle) or $10^5$ (right) TCID$_{50}$ of SARS-CoV-2. (C) Frequencies of CD8$^+$ T cells targeting S, N, and M & E in submandibular lymph nodes obtained at necropsy in macaques N011, N013, N021, and N022. Samples were unavailable for analysis in macaque N012.

(1.4 x $10^8$ RNA copies) of SARS-CoV-2, viral RNA copies in nasopharyngeal swabs on day 2 were comparable (N012) to or greater (N013, N022, D024, and D025) than the total viral RNA copies in the inoculum, confirming viral replication in macaques even with $10^5$ TCID$_{50}$. The three macaques inoculated with $10^6$ TCID$_{50}$ showed similar levels of viral RNA in nasopharyngeal swabs on days 2 and 5 compared to the five macaques with $10^5$ TCID$_{50}$. All the eight macaques intranasally inoculated with $10^6$ or $10^5$ TCID$_{50}$ of SARS-CoV-2 developed efficient anti-SARS-CoV-2 NAb responses, and the five anti-CD8 antibody-untreated macaques induced SARS-CoV-2-specific CD8$^+$ T-cell responses. Taken together, our results show that intranasal inoculation of cynomolgus macaques with $10^6$ or $10^5$ TCID$_{50}$ of SARS-CoV-2 results in viral replication in the pharyngeal mucosa. Containment of viral replication in the

pharyngeal mucosa would be important for the control of further viral transmission as well as disease progression.

Our cynomolgus macaque model of intranasal but not intratracheal SARS-CoV-2 inoculation is considered to represent asymptomatic or mild COVID-19. However, histopathological analysis of the lung detected pulmonary inflammation in one animal (N021) on day 14 post-infection (S2 Fig), suggesting the potential of intranasal SARS-CoV-2 inoculation to induce moderate pulmonary diseases. Other animals may also have developed mild pulmonary inflammation detectable on day 14, which was resolved by day 21. Macaque N013 showed a unique phenotype with undetectable viral RNAs in swabs after day 14 (Fig 1D) but relatively higher levels of viral RNA in pharyngeal mucosa and submandibular lymph nodes on day 21 post-infection (Table 3). Virus-specific CD8$^+$ T-cell responses were undetectable in PBMCs but efficiently detected in the submandibular lymph nodes on day 21 (Fig 5), suggesting localized virus replication in the pharyngeal mucosa.

The sample size used in this study is relatively limited (three anti-CD8 antibody treated animals and five untreated controls). However, these three animals exhibited similar levels of pharyngeal viral loads before the anti-CD8 antibody treatment, and again showed similar levels of viral loads after CD8$^+$ cell depletion. Neither enhancement of viral replication nor delay in viral control was observed. Regarding the five anti-CD8 antibody untreated macaques, SARS-CoV-2-specific CD8$^+$ T-cell responses were detected in all, although the magnitudes and kinetics of these responses were different. Therefore, this study provides sufficient evidence for our conclusion.

In summary, the present study showed that subacute viral replication can be controlled even in the absence of CD8$^+$ T cells in primary SARS-CoV-2 infection. CD8$^+$ T-cell responses may contribute to viral control in SARS-CoV-2 infection, but our results suggest that CD8$^+$ T-cell dysfunction does not solely lead to viral control failure or disease progression.

## Materials and methods

### Ethics statement

Animal experiments were performed in the National Institute of Infectious Diseases (NIID) after approval by the Committee on the Ethics of Animal Experiments in NIID (permission number: 520001) under the guidelines for animal experiments in accordance with the Guidelines for Proper Conduct of Animal Experiments established by the Science Council of Japan (http://www.scj.go.jp/ja/info/kohyo/pdf/kohyo-20-k16-2e.pdf). The experiments were in accordance with the "Weatherall report for the use of non-human primates in research" recommendations (https://royalsociety.org/topics-policy/publications/2006/weatherall-report/). Each macaque was housed in a separate cage and received standard primate feed and fresh fruit daily. Virus inoculation, blood collection, nasopharyngeal and throat swab collection, and anti-CD8 antibody treatment were performed under ketamine anesthesia. Macaques were euthanized by whole blood collection under deep anesthesia on day 14 or 21 post-infection.

### Animal experiments

SARS-CoV-2 wk-521 strain [39] (2019-nCoV/Japan/TY/WK-521/2020, GenBank Accession LC522975) was expanded in Vero E6/TMPRSS2 cells [39] and harvested to prepare a virus inoculum stock. Virus infectivity was assessed by detection of cytopathic effect (CPE) on Vero E6/TMPRSS2 cells and determination of endpoint titers. Nine cynomolgus macaques (*Macaca fascicularis*, 3–6 years old) were intranasally inoculated with the same stock of SARS-CoV-2 wk-521 at a dose of $10^6$ (exactly 7.5 x $10^5$) TCID$_{50}$ (1.4 x $10^9$ RNA copies) (n = 3), $10^5$ (exactly 7.5 x $10^4$) TCID$_{50}$ (n = 5), or $10^4$ (exactly 7.5 x $10^3$) TCID$_{50}$ (n = 1) (Table 1). Three (Group D)

of the nine macaques were intravenously administered with 5 mg/kg body weight of anti-CD8α antibody clone MT807 (NIH Nonhuman Primate Reagent Resource) on days 5 and 7 post-infection. Body temperature was measured with a small implantable thermo logger (DST micro-T; Star-Oddi) that was set intraperitoneally under ketamine anesthesia at least five days prior to virus inoculation. Macaques were euthanized and subjected to necropsy on day 14 or 21 post-infection (Table 1).

## Detection of SARS-CoV-2 RNAs

Swab RNA was extracted from 0.2 ml of swab solutions (1ml of DMEM with 2% fetal bovine serum [Cytiva]) using QIAamp Viral RNA Minikit (QIAGEN) and subjected to real-time RT-PCR for viral RNA quantitation [40] using QuantiTect Probe RT-PCR Kit (Qiagen) and QuantStudio 5 (Thermo Fisher Scientific). Swab RNAs were also subjected to real-time RT-PCR for measurement of viral subgenomic RNA (sgRNA) levels [32,33,41] using the following primers: SARS2-LeaderF60 (5'-CGATCTCTTGTAGATCTGTTCTCT-3'), SARS2-N28354R (5'-TCTGAGGGTCCACCAAACGT-3'), and SARS2-N28313Fam (FAM-TCAGC GAAATGCACCCCGCA-TAMRA). Tissue RNAs were extracted from homogenized tissues by using TRIzol Plus RNA Purification Kit (Thermo Fisher Scientific) with phenol-chloroform extraction and subjected to real-time RT-PCR for detection of viral RNAs.

## Virus recovery from swabs

Vero E6/TMPRSS2 cells in 96-well plates were added with 10-fold serially diluted swab solutions and cultured for 4 days without medium change. Virus recovery was assessed by detection of CPE and determination of endpoint titers. Swab samples with virus titers greater than 1 x $10^2$ $TCID_{50}$/swab were considered positive.

## Analysis of anti-SARS-CoV-2 NAb responses

Plasma samples were heat inactivated for 30 min at 56°C. Serial two-fold dilutions of heat-inactivated plasma were tested in quadruplicate. In each mixture for quadruplicate testing, 40 μl of diluted plasma were incubated with 40 μl of 80 $TCID_{50}$ SARS-CoV-2 wk-521. After incubation for 45 min at room temperature, 20 μl of the mixture was added to each of four wells (1 x $10^4$ Vero cells/well) in a 96-well plate. Three days later, virus infectivity was assessed by detection of CPE to determine the endpoint titers. The lower limit of detection was 1:10.

## Analysis of cell surface markers

Whole blood samples were treated with Lysing Solution (BD) and subjected to surface staining using anti-CD3 APC-Cy7 (SP34-2; BD), anti-CD4 FITC (M-T477; BD), anti-CD8 PerCP (SK1; BD), and anti-CD20 PE (2H7; BD) antibodies. Alternatively, whole blood samples from anti-CD8 antibody-treated animals were stained with anti-CD3 APC-Cy7, anti-CD4 PerCP (L200; BD), anti-CD8 FITC (DK25; FUJIFILM), and anti-CD20 PE. Stained cells were analyzed by BD FACS Canto II.

## Analysis of SARS-CoV-2 antigen-specific CD8+ T-cell responses

Virus-specific CD8+ T-cell frequencies were measured by flow cytometric analysis of gamma interferon (IFN-γ) induction after specific stimulation as described previously [42]. PBMCs were prepared from whole blood by density gradient centrifugation using Ficoll-Paque PLUS (Cytiva). Lymph node-derived lymphocytes were prepared from minced lymph nodes by density gradient centrifugation using Ficoll-Paque PLUS. Cells were pulsed and cocultured with

peptide pools (at a final concentration of more than 0.1 μM for each peptide) using panels of overlapping peptides spanning the SARS-CoV-2 S, N, M, and E amino acid sequences (PM-WCPV-S-1, PM-WCPV-NCAP-1, PM-WCPV-VME-1, and PM-WCPV-VEMP-1; JPT Peptide Technologies) in the presence of GolgiStop (monensin, BD), 1 μg/ml of anti-CD28 (CD28.2, BD) and 1 μg/ml anti-CD49d (9F10, BD) for 6 hours. Intracellular IFN-γ staining was performed with a CytofixCytoperm kit (BD) and anti-CD3 APC-Cy7, anti-CD4 FITC, anti-CD8 PerCP, and anti-IFN-γ PE (4S.B3; BioLegend). Stained cells were analyzed by BD FACS Lyric. A representative gating schema for flow cytometric analysis is shown in Fig 5A. Specific T-cell frequencies were calculated by subtracting nonspecific IFN-γ$^+$ T-cell frequencies from those after peptide-specific stimulation. Specific T-cell frequencies less than 0.03% of CD8$^+$ T cells were considered negative.

## Statistical analysis

Statistical analyses were performed using Prism software (GraphPad Software, Inc.) with significance set at $p$ values of $< 0.05$. Comparisons were performed by Mann-Whitney U test.

## Supporting information

**S1 Fig. Changes in body temperatures pre- and post-infection in macaques.**
(TIF)

**S2 Fig. Histopathology of the lung in macaque N021.** Representative histopathology with hematoxylin and eosin staining (H&E) of the lung obtained from macaque N021 at autopsy on day 14 post-infection, indicating mild or moderate pulmonary inflammation. Infiltration of mononuclear cells were observed around blood vessels and bronchiole (upper left panel). Lymphocytes, eosinophils, and macrophages were observed in pulmonary alveoli (upper right and lower panels).
(TIF)

**S3 Fig. CD8$^+$ T-cell frequencies in the submandibular lymph nodes at necropsy.**
CD3$^+$CD8$^+$ cell frequencies in the submandibular lymph nodes obtained at autopsy are shown.
(TIF)

## Acknowledgments

We thank S. Matsuyama and M. Takeda for providing SARS-CoV-2 wk-521 and Vero E6/TMPRSS2 cells, H. Ohashi and K. Watashi for providing Vero cells, and S. Fukushi for his help. We also thank M. de Souza for editing the paper.

## Author Contributions

**Conceptualization:** Takushi Nomura, Tetsuro Matano.

**Data curation:** Takushi Nomura, Shigeyoshi Harada, Hiroshi Ishii, Tetsuro Matano.

**Formal analysis:** Takushi Nomura, Hiroyuki Yamamoto, Shigeyoshi Harada, Hiroshi Ishii, Ai Kawana-Tachikawa, Noriyo Nagata, Naoko Iwata-Yoshikawa, Eun-Sil Park, Tetsuro Matano.

**Funding acquisition:** Ai Kawana-Tachikawa, Tetsuro Matano.

**Investigation:** Takushi Nomura, Hiroyuki Yamamoto, Masako Nishizawa, Trang Thi Thu Hau, Shigeyoshi Harada, Hiroshi Ishii, Sayuri Seki, Midori Nakamura-Hoshi, Midori Okazaki, Sachie Daigen, Ai Kawana-Tachikawa, Noriyo Nagata, Naoko Iwata-Yoshikawa, Nozomi Shiwa, Tadaki Suzuki, Eun-Sil Park, Yuriko Suzaki, Yasushi Ami.

**Methodology:** Takushi Nomura, Shigeyoshi Harada, Hiroshi Ishii, Ai Kawana-Tachikawa, Noriyo Nagata, Naoko Iwata-Yoshikawa, Shun Iida, Harutaka Katano, Eun-Sil Park, Ken Maeda, Yasushi Ami.

**Project administration:** Takushi Nomura, Yasushi Ami, Tetsuro Matano.

**Supervision:** Tetsuro Matano.

**Validation:** Takushi Nomura, Shigeyoshi Harada, Hiroshi Ishii, Tetsuro Matano.

**Visualization:** Takushi Nomura, Tetsuro Matano.

**Writing – original draft:** Takushi Nomura, Tetsuro Matano.

**Writing – review & editing:** Takushi Nomura, Hiroyuki Yamamoto, Shigeyoshi Harada, Hiroshi Ishii, Ai Kawana-Tachikawa, Noriyo Nagata, Tadaki Suzuki, Ken Maeda, Yasushi Ami, Tetsuro Matano.

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
