## [Decision Letter · Decision Letter 0]

28 Jun 2021

Dear Prof. Matano,

Thank you very much for submitting your manuscript "Subacute SARS-CoV-2 replication can be controlled in the absence of CD8+ T cells in cynomolgus macaques" for consideration at PLOS Pathogens. As with all papers reviewed by the journal, your manuscript was reviewed by members of the editorial board and by several independent reviewers. The reviewers appreciated the attention to an important topic. Based on the reviews, we are likely to accept this manuscript for publication, providing that you modify the manuscript according to the review recommendations.

All three reviewers thought this was a timely and significant study. While the number of animals used is limited, the authors have acknowledged this limitations. I ask that the authors address the minor comments raised by Reviewers #2 and #3. These can be addressed editorially in the text of the revised manuscript. 

Sincerely,

Benhur Lee

Section Editor

PLOS Pathogens

Benhur Lee

Section Editor

PLOS Pathogens

Kasturi Haldar

Editor-in-Chief

PLOS Pathogens

orcid.org/0000-0001-5065-158X

Michael Malim

Editor-in-Chief

PLOS Pathogens

orcid.org/0000-0002-7699-2064

Reviewer Comments (if any, and for reference):

Reviewer's Responses to Questions

**Part I - Summary**

Reviewer #1: Nomura et al CD8 deplete 3 macaques 5 days after SARS-CoV-2 infection and show the profile of resolution of infection is similar in these 3 macaques to 5 controls. The work is well done and of interest to the COVID field grappling with a role if any for CD8 T cells in control of COVID.

Comments

Although acknowledged by the authors, the numbers of macaques and groups are very limited. I felt a more definitive data set would have perhaps considered a B cell depletion positive control.

Only one time point of depletion was studied. I felt that day 5 was a reasonable stab at showing the importance of control, but it remains possible it was too late.

The role of CD8 T cells, if any, may be more important in prevention of severe disease, which macaques rarely suffer from. An alternate role may be in control of reinfection, as suggested by Barouch et al (albeit with limited data).

Reviewer #2: This study uses a macaque model with intranasal SARS-CoV-2 exposure to assess the role of CD8+ T-cells in viral control. The results demonstrate that CD8+ T cell depletion does not impact viral clearance during SARS-CoV-2 primary infection. The manuscript is well written, but could be improved by addressing some concerns related to study design and interpretation of the results.

1. The investigators should clarify the apparent timing issue where viral clearance occurs predominantly by 10 days post-infection, while induction of systemic CD8+ T cell responses occurs by 14 days post-infection. Their proposed temporal sequence of T-cell and B-cell activity in the primary infection should be provided in a diagram to illustrate the possible relationship with viral clearance.

2. The investigators should comment on whether there is actual depletion of CD8+ T cells in the tissues, especially at sites of SARS-CoV-2 replication. Looking at tissue specific CD8+T cells responses would be more informative considering where the virus replicates. This is demonstrated by authors own data on animal N013 that has no detectable systemic CD8+T cell responses but has responses as well as viral replication in submandibular lymph nodes.

3. While the investigators find systemic SARS-CoV-2 specific CD8+ T cell responses in untreated macaques, the responses have a wide range. Even in the 5 animals examined, a wide range of responses can be noticed from no response (N013) to some response at d14. Having only 3 animals in the active arm and a wide range of responses in the control animals are two important limitations impacting each other in the interpretation of the results that the investigators should comment on.

4. Since cellular immunity may play a more important role in clearing/preventing subsequent infections and controlling outcome in severe/uncontrolled infections, the investigators should comment on the need for further studies using a reinfection model to assess the role of CD8+ T cell responses in viral clearance.

5. Statements in lines 203-205 and 250-252 need to be adjusted to clearly state that the observations are only relevant for primary, controlled SARS-CoV-2 infections.

Reviewer #3: Nomura and colleagues assess the impact of CD8 T cell depletion on the control of subacute SARS-CoV-2 replication in a small cohort of cynomolgus macaques. They report that when CD8 depletion is performed 5-7 days after SARS-CoV-2 infection, there is no demonstrable impact on viral replication compared to control animals. They correctly surmise in the discussion that while this does not dispute a potential role for vaccine-induced CD8 T cells in protection from infection, the data suggest that CD8 T cells are not strictly required to control low-level viral replication in the pharyngeal mucosa.

The role for T cells in determining COVID-19 severity or vaccine-induced protection is a topic of significant interest, which can be challenging to conclusively assess in human cohorts. While the authors note the limitations of the current study, including the sample size, non-human primate studies are a valuable pre-clinical model to assess the contributions of various immune cells to viral control. The manuscript is well-written and presents the data and conclusions in a clear and concise manner. I have no substantial concerns about the study, but some additional discussion points would enhance its message (see below).

**Part II – Major Issues: Key Experiments Required for Acceptance**

Reviewer #1: see above

Reviewer #2: No additional experiments required

Reviewer #3: None

**Part III – Minor Issues: Editorial and Data Presentation Modifications**

Reviewer #1: see above

Reviewer #2: (No Response)

Reviewer #3: One of the major considerations for the role of CD8 T cells in controlling viral replication during primary infection versus the contribution of T cells to protection from re-infection includes the kinetics of the T cell response. Based on the data presented in Figure 5, it is relatively clear that primary, subacute infection is unlikely to induce detectable CD8 responses prior to day 14 in most animals. Given how early viral replication peaks in this animal model, it seems likely that the primary CD8 response simply occurs too late to contribute to viral control. I think the discussion could more explicitly point this out, and perhaps report what is known about similar kinetics in humans.

PLOS authors have the option to publish the peer review history of their article (what does this mean?). If published, this will include your full peer review and any attached files.

Reviewer #1: No

Reviewer #2: No

Reviewer #3: No

Figure Files:

Data Requirements:

Reproducibility:

References:

---

## [Editor Report · Decision Letter 1]

6 Jul 2021

Dear Prof. Matano,

We are pleased to inform you that your manuscript 'Subacute SARS-CoV-2 replication can be controlled in the absence of CD8+ T cells in cynomolgus macaques' has been provisionally accepted for publication in PLOS Pathogens.

Best regards,

Benhur Lee

Section Editor

PLOS Pathogens

Benhur Lee

Section Editor

PLOS Pathogens

Kasturi Haldar

Editor-in-Chief

PLOS Pathogens

orcid.org/0000-0001-5065-158X

Michael Malim

Editor-in-Chief

PLOS Pathogens

orcid.org/0000-0002-7699-2064
---

## [Editor Report · Acceptance letter]

16 Jul 2021

Dear Prof. Matano,

We are delighted to inform you that your manuscript, "Subacute SARS-CoV-2 replication can be controlled in the absence of CD8+ T cells in cynomolgus macaques," has been formally accepted for publication in PLOS Pathogens.

Best regards,

Kasturi Haldar

Editor-in-Chief

PLOS Pathogens

orcid.org/0000-0001-5065-158X

Michael Malim

Editor-in-Chief

PLOS Pathogens

orcid.org/0000-0002-7699-2064